# Free d-Amino Acids in Salivary Gland in Rat

**DOI:** 10.3390/biology11030390

**Published:** 2022-03-02

**Authors:** Masanobu Yoshikawa, Takugi Kan, Kosuke Shirose, Mariko Watanabe, Mitsumasa Matsuda, Kenji Ito, Mitsuru Kawaguchi

**Affiliations:** 1Department of Clinical Pharmacology, School of Medicine, Tokai University, Isehara 259-1193, Japan; 2Department of Anesthesiology, School of Medicine, Tokai University, Isehara 259-1193, Japan; tkan@is.icc.u-tokai.ac.jp (T.K.); sk327798@tsc.u-tokai.ac.jp (K.S.); 9j236559@is.icc.u-tokai.ac.jp (M.W.); anes-mm@is.icc.u-tokai.ac.jp (M.M.); itokenji@is.icc.u-tokai.ac.jp (K.I.); 3Tokyo Dental College, Tokyo 101-0061, Japan; kawaguti@tdc.ac.jp

**Keywords:** d-amino acid, salivary gland, NMDA receptor, serine racemase, d-amino acid oxidase, d-aspartate oxidase

## Abstract

**Simple Summary:**

Parotid, submandibular, and sublingual glands in rat were found to contain high concentrations of d-aspartic acid and low ones of d-serine and d-alanine. In addition to d-amino acid oxidase and d-aspartate oxidase, serine racemase was also detected in all three of these major salivary glands, as were *N*-methyl-d-aspartic acid receptor subunits NR1 and NR2D, but not NR2A, NR2B, or NR2C.

**Abstract:**

Free d-amino acids, which are enantiomers of l-amino acids, are found in mammals, including humans, and play an important role in a range of physiological functions in the central nervous system and peripheral tissues. Several d-amino acids have been observed in saliva, but their origin and the enzymes involved in their metabolism and catabolism remain to be clarified. In the present study, large amounts of d-aspartic acid and small amounts of d-serine and d-alanine were detected in all three major salivary glands in rat. No other d-enantiomers were detected. Protein expression of d-amino acid oxidase and d-aspartate oxidase, the enzymes responsible for the oxidative deamination of neutral and dicarboxylic d-amino acids, respectively, were detected in all three types of salivary gland. Furthermore, protein expression of the d-serine metabolic enzyme, serine racemase, in parotid glands amounted to approximately 40% of that observed in the cerebral cortex. The *N*-methyl-d-aspartic acid subunit proteins NR1 and NR2D were detected in all three major salivary glands. The results of the present study suggest that d-amino acids play a physiological role in a range of endocrine and exocrine function in salivary glands.

## 1. Introduction

Free d-amino acids, which are enantiomers of l-amino acids [1], are found in mammals, including humans [2,3,4], and play an important role in a range of physiological functions [5]. Among free d-amino acids in mammals, d-serine and d-aspartic acid have been studied extensively. 

d-Serine is known to be widely present in central nervous tissues, peripheral tissues, and body fluids [2,6,7]. d-Serine, in particular, is abundant in the mammalian forebrain, where it is involved in higher brain function during the entire postnatal period, acting as an endogenous and obligatory coagonist at the glycine site of the *N*-methyl-d-aspartic acid (NMDA) receptor [8,9]. Substantial amounts of d-serine were detected in extracellular space in the forebrain, where NMDA receptors are abundant [10], indicating that it is involved in glutamatergic neurotransmission via these receptors [11,12]. In contrast to in central nervous tissues, only trace levels of d-serine were detected in postnatal peripheral tissues [13].

d-Aspartic acid was the first free d-amino acid to be identified in mammals [14], and has been shown to demonstrate affinity for the glutamic acid binding site on the NR2 subunit of NMDA receptors [15]. Although the role of d-aspartic acid in the central nervous system remains unclear due to its presence in only low levels postnatally, it is expressed in a developmentally relevant manner in various endocrine and neuroendocrine organs [13,16,17]. d-Aspartic acid is considered to regulate the synthesis and secretion of hormones and proliferation and differentiation of endocrine cells in mammals [18,19,20,21,22,23,24]. Maximal d-aspartic acid content corresponds well with the period of morphological and functional maturation of various organs, suggesting its involvement in regulating the synthesis and secretion of hormones for these sites [25,26,27].

Three main sources have been posited for d-amino acids in mammals. The first, which seems to be restricted to d-serine and d-aspartic acid, is tissue; these are synthesized by the racemization of their corresponding l-amino acids. In mammals, a serine racemase has been identified which converts l-serine to d-serine [28]. On the other hand, no aspartate racemase has been identified in mammals so far, although serine racemase is suggested to act as a biosynthetic enzyme of d-aspartic acid in some tissue [29]. The second is diet. Fermented foods are enriched in d-amino acids, including d-alanine, d-aspartic acid, d-glutamic acid, and d-proline [30]. The third is intestinal bacteria, which account for a range of such amino acids, including d-alanine, d-aspartic acid, d-glutamic acid, and d-proline [31].

d-Serine and d-aspartic acid are catabolized by d-amino acid oxidase (DAO, EC 1.4.3.3) and d-aspartate oxidase (DDO, EC 1.4.3.1), respectively, in mammals. d-Amino acid oxidase catalyzes oxidative deamination of neutral and basic d-amino acids in mammals, which generate alpha-keto acids, ammonium ion, and hydrogen peroxide [32]. Among d-amino acids, d-alanine and d-serine are regarded as the primary substrates of mammalian DAO, as they show a significant increase in rodents lacking DAO in comparison with other d-amino acids [33,34]. d-Aspartate oxidase catalyzes the oxidative deamination of acidic d-amino acids which possess a dicarboxylic group, such as d-aspartic acid and d-glutamic acid, in mammals [35]. The decrease in d-aspartic acid levels observed in the brain during the postnatal period is the result of increased expression of DDO [36].

d-Alanine is predominantly distributed in endocrine tissues in mammals [37,38,39], acting as a coagonist at the glycine site of the NMDA receptor. The origins of d-alanine may be found in intestinal bacteria and/or dietary intake. d-Alanine is catabolized by DAO, in the same way as d-serine [34,40].

The NMDA receptor heterotetrameric transmembrane ion channel is comprised of four subunits: NR1, NR2A-D, and NR3A and B. The NR1 subunit is essential for receptor assembly and functional activity. These four NR2 subunits, primary determinants of the functional heterogeneity of the receptor, show markedly different temporal and spatial expression [41,42]. Activation of NMDA receptors requires binding of glutamic acid to NR2 subunits along with a co-agonist, glycine or d-serine, to the NR1 subunit [8,9]. While the NMDA receptor was originally identified in the central nervous system, it is also expressed in a wide range of peripheral tissues [32,43,44,45,46,47,48,49].

In the present study, the amounts of the d- and l-enantiomers of 21 amino acids and glycine in the three major salivary glands was determined comprehensively in rat using two-dimensional high performance liquid chromatography (2D-HPLC). To investigate the enzymes involved in the metabolism and catabolism of d-amino acid in rat salivary gland, gene and protein expression of serine racemase, DAO and DDO was also evaluated. In addition, gene and protein expression of NMDA receptor subunits in rat salivary glands were determined.

## 2. Materials and Methods

### 2.1. Animals

Male Wistar rats (7–8 weeks old, 230–250 g each, n = 25; Clea Japan Inc., Tokyo, Japan) were housed in an air-conditioned room (temperature 24–26 °C, humidity 50–60%) under a 12 h light/dark cycle (lights on: 7:00); food (CE-2; Clea Japan Inc., Tokyo, Japan) and water were freely available. Rats were allowed 1 week to adapt well to the novel laboratory environment.

### 2.2. Chemicals

The amino acids were obtained from Sigma (St. Louis, MO, USA), Tokyo Kasei Kogyo Co. (Tokyo, Japan) and FUJIFILM Wako Chemical Co. (Osaka, Japan). 4-Fluoro-7-nitro-2,1,3-benzoxadiazole (NBD-F) were obtained from Tokyo Kasei Kogyo Co. (Tokyo, Japan). Primers were obtained from Sigma Japan (Tokyo, Japan). Unless otherwise indicated, all chemicals were purchased from FUJIFILM Wako Chemical Co. (Osaka, Japan).

### 2.3. Sample Tissue Preparation

Three rats were euthanized by exsanguination via the abdominal aorta under anesthesia with pentobarbital (50 mg/kg, intraperitoneal administration). The three major salivary glands were quickly excised and stored at −80 °C until use. The tissues were homogenized at 3500 rpm for 2 min in water (20× volume of tissue wet weight) at 4 °C using Micro Smash (MS-100R, TOMY Seiko Co., Tokyo, Japan). The homogenates were centrifuged at 12,000× *g* for 10 min. A total of 200 µL methanol was added to 50 µL supernatant and centrifuged at 12,000× *g* for 10 min. Fifty µL supernatant was evaporated to dryness under reduced pressure at 40 °C. Twenty µL of 200 mM sodium borate buffer (pH 8.0) and 5 µL of 40 mM NBD-F in dry acetonitrile was added to the residue and heated at 60 °C for 2 min. To terminate the derivatization reaction, 75 µL of 2% (*v*/*v*) trifluoroacetic acid in water was added. Two µL reaction mixture was then injected into the 2D-HPLC system. The extraction of amino acids from salivary gland tissue and 2D-HPLC experiments in this study were performed at Shiseido Co. (Tokyo, Japan).

### 2.4. Determination of Amino Acid Enantiomers by 2D-HPLC

The enantiomers of the amino acids were quantified with a 2D-HPLC system (NANOSPACE SI-2 series, Shiseido, Tokyo, Japan) using a method described previously [34,50].

### 2.5. Real Time-Quantitative Reverse Transcriptase-Polymerase Chain Reaction (RT-PCR)

Total RNA was extracted according to the manufacturers’ protocol using the SV total RNA extraction Kit (Promega Co., Madison, WI, USA). Gene expression of serine racemase (GenBank accession number NM_198757.2) and DAO (accession number NM_053626.1) was determined using the glyceraldehyde-3-phosphate dehydrogenase (GAPDH) (GenBank accession number NM_017008) gene as an internal control, as described previously [51,52]. Gene expression of DDO was determined using primers specific to DDO mRNA (accession number NM_001109465.2) (upper primer, AAC CCT GGG AGG GAG TAG AC; lower primer, TTA TGT CGC AGG CTC TGT; product size, 122 base pairs). Gene expression of each subunit of the NMDA receptor was determined using primers specific to NR1 subunit mRNA (accession number NM_017010.2) (upper primer, ACA AGC GAC ACA AGG ATG C; lower primer, GGG CTC TGC TCT ACC ACT CTT; product size, 107 base pairs), NR2A subunit (NR2A) mRNA (NM_012573.3) (upper primer, CAA CCT GGC TGC CTT CAT; lower primer, AGA ATG GTC ATG AGG TCT CTG GAA C; product size, 91 base pairs), NMDA receptor 2B subunit (NR2B) mRNA (NM_012574.1) (upper primer, TCC TGC AGC TGT TTG GAG AT; lower primer, GCT GCT CAT CAC CTC ATT CTT; product size, 95 base pairs), NR2C subunit (NR2C) mRNA (NM_012575.3) (upper primer, GGC ACT CCT GCA ACT TCT G; lower primer, GTT CTG GCA GAT CCC TGA GA; product size, 76 base pairs) and NR2D subunit (NR2D) mRNA (NM_022797.2) (upper primer, GCA GCA ATG GCA CTG TGT; lower primer, ACA TCA TCA CCC AGA CAG CA; product size, 69 base pairs). The cDNA was amplified using the DyNAmo SYBER green qPCR Kit (ThermoFisher Scientific; Waltham, MA, USA) and the DNA Engine Opticon 2 System (Bio-Rad Laboratories; Hercules, CA, USA), running 40 cycles of the following protocol: 10 min predenaturation at 95 °C, 15 s annealing at 64.4 °C for DDO, NR1, NR2C, and NR2D or annealing at 61 °C for NR2A and NR2B, followed by a 20 s extension at 72 °C.

### 2.6. Capillary Electrophoresis-Based Immunodetection Assay (Simple Western)

Total protein was extracted from rat salivary glands, cerebral cortex and cerebellum using a method described previously with modifications [53,54]. Briefly, the tissues were separately homogenized on ice in protein lysis buffer (50 mM Tris-HCl, pH 7.6, 150 mM NaCl, 1% Nonidet P40, 0.5% sodium deoxycholate, 1% sodium dodecyl sulfate; RIPA buffer, Nacalai, Kyoto, Japan) supplemented with protease inhibitor cocktail (cOmplete Tablets, EDTA-free, Roche Diagnostics, Mannheim, Germany) and phosphatase inhibitor cocktail (PhosSTOP, EDTA-free, Roche Diagnostics, Mannheim, Germany). The homogenate was then centrifuged at 12,000× *g* for 20 min at 4 °C. Protein concentration in the supernatant solution was determined using a DC Protein Assay Kit (Bio-Rad Laboratory, Hercules, CA, USA). Six µg extracted protein was loaded into the capillary, except for with detection of GAPDH as an internal control, for which 1 µg was applied. The samples were prepared according to the manufacturer’s protocol using a Protein Simple Separation Module (Protein Simple, Santa Clara, CA, USA). Separation and detection of the target protein were performed with Wes (Protein Simple, Santa Clara, CA, USA) using the following antibodies: anti-serine racemase (1:50 dilution, ab182217, Abcam, Cambridge, UK), anti-DAO (1:50 dilution, sc-398757, Santa Cruz Biotechnology, Dallas, TX, USA), anti-DDO (1:50 dilution, 13682-AP-1, Proteintech, Rosemont, IL, USA), anti-NR1 (1:50 dilution, sc-518053, Santa Cruz Biotechnology, Dallas, TX, USA), anti-NR2A (1:50 dilution, sc-515148, Santa Cruz Biotechnology, Dallas, TX, USA), anti-NR2B (1:50 dilution, sc-365597, Santa Cruz Biotechnology, Dallas, TX, USA), anti-NR2C (1:50 dilution, 600-401-D94, Rockland, Limerick, PA, USA), anti-NR2D (1:50 dilution, sc-17822, Santa Cruz Biotechnology, Dallas, TX, USA) and anti-GAPDH (1:300 dilution, G9545, Sigma, St. Lous, MO, USA). To quantitate and validate the amount of the target protein in each sample, detected chemiluminescent signals were analyzed using a software package (Compass, Protein Simple, Santa Clara, CA, USA). The quantitative data were obtained from the area of the peak formed at the expected molecular weight of the protein. The area values were calculated as intensity bands for each protein in each tissue and normalized to the housekeeping gene GAPDH in each tissue.

### 2.7. Statistical Analyses

The results are presented as the mean and standard deviation (SD). All statistical analysis was performed by software package Prism 6.0c (GraphPad Software Inc., San Diego, CA, USA). The Mann–Whitney test was used for comparisons between two groups. Dunn’s multiple comparison test was used to determine significance in each group when a significant difference among groups was obtained by Kruskal–Wallis tests for more than two groups. A *p*-value of less than 0.05 was considered to indicate statistical significance.

## 3. Results

### 3.1. Determination of Amino Acid Enantiomers in Rat Salivary Glands Tissues Using 2D-HPLC

A large amount of d-aspartic acid (104.7–174.3 nmol/g) was detected together with small amounts of d-serine (3.8–4.9 nmol/g) and d-alanine (11.6–14.1 nmol/g) in the three major salivary glands in rat according to the 2D-HPLC system (Table 1). No other d-enantiomers were detected.

### 3.2. Gene Expression of Serine Racemase, DAO, and DDO in Three Major Salivary Glands

Levels of serine racemase mRNA in the parotid, submandibular, and sublingual glands amounted to approximately 400%, 180%, and 120%, respectively, of those observed in the cerebral cortex and cerebellum (Figure 1A). Levels of DAO mRNA in the parotid, submandibular, and sublingual glands amounted to approximately 70%, 30%, and 15%, respectively, of that observed in the cerebellum (Figure 1B). Levels of DDO mRNA in the parotid, submandibular, and sublingual glands amounted to approximately 75%, 20%, and 15%, respectively, of that observed in the cerebral cortex (Figure 1C).

### 3.3. Protein Expression of Serine Racemase, DAO, and DDO in Three Major Salivary Glands

Protein expression of serine racemase in the parotid, submandibular, and sublingual glands amounted to approximately 40%, 20%, and 15%, respectively, of that observed in the cerebral cortex (Figure 2B). Protein expression of DAO in the parotid, submandibular, and sublingual glands amounted to approximately 30%, 15%, and 20%, respectively, of that observed in the cerebellum (Figure 2B). Protein expression of DDO in the parotid, submandibular, and sublingual glands amounted to approximately 7%, 8%, and 5%, respectively, of that observed in the cerebral cortex (Figure 2B). Compared to in the controls (cerebral cortex or cerebellum), protein levels of serine racemase, DAO, and DDO in all three salivary glands were relatively low; that is, they differed from their mRNA expression levels.

### 3.4. Gene Expression of NMDA Receptor Subunits in Three Major Salivary Glands

Figure 3 shows the levels of NMDA receptor subunit mRNA in a mixture of three salivary glands and whole brain. In salivary glands, NR2D mRNA was detected to some extent, but NR2A, NR2B, and NR2C at only trace levels. Levels of NR1 and NR2D mRNA in the mixture of three salivary glands were nearly equal. Levels of NR1 mRNA in the parotid, submandibular, and sublingual glands amounted to approximately 6%, 1%, and 5%, respectively, of that observed in the cerebral cortex (Figure 4). Levels of NR1 and NR2D mRNA in the parotid gland were higher than those observed in the submandibular or sublingual gland. Levels of NR1 and NR2D mRNA in the parotid, submandibular, and sublingual gland were nearly equal.

### 3.5. Protein Expression of NR1 and NR2D in Three Major Salivary Glands

Protein expression of NR1 in the parotid, submandibular, and sublingual glands amounted to approximately 2%, 1%, and 1%, respectively, of that observed in the cerebral cortex (Figure 5). Protein expression of NR2D mRNA in the parotid, submandibular, and sublingual glands amounted to approximately 40%, 30%, and 20%, respectively, of that observed in the cerebral cortex (Figure 5). The ratio of NR1 to NR2D protein expression was almost equal (1:1) in the parotid, submandibular, and sublingual glands.

## 4. Discussion

Three d-amino acids—d-aspartic acid, d-serine, and d-alanine—were detected in parotid, submandibular, and sublingual gland in rat. No other d-enantiomers were detected. To the best of our knowledge, this is the first study to comprehensively determine the amounts of these d-amino acids in salivary glands. Large amounts of d-aspartic acid, in particular, were detected in all three major salivary glands. An earlier study found a high amounts and %D, calculated as D/(D + L) × 100, of d-aspartic acid in the pineal gland (1131.0 ± 232.6 nmol/g wet tissue, %D = 59%), pituitary gland (153.3 ± 24.3 nmol/g wet tissue, %D = 5%) and adrenal gland (21.6 ± 11.1 nmol/g wet nmol/g, %D = 3%) in rat [26]. In the present study, the amounts and %D value of d-aspartic acid detected in the three major salivary glands were equal to, or higher than, those found in other endocrine organs [26], except for the highest %D value, which was observed in the pineal gland. In light of the fact that d-aspartic acid regulates synthesis and secretion in various types of endocrine tissue [19,20,21], the present finding provides further support for the view that that salivary glands are not just exocrine, but also endocrine organs as well [55,56,57,58]. Salivary glands synthesize and release several peptides and hormones into the blood, including glucagon [59], epidermal growth factor [60], nerve growth factor [61], sialorphin [62,63], glucocorticoids, and sex steroid hormones [64]. The results of the present study suggest that d-aspartic acid participates in the regulation of the function of the salivary glands as an endocrine organ.

High levels of d-aspartic acid are found in the brain and peripheral tissues during the critical period when organs are maturing morphologically and functionally, suggesting its involvement in the regulation of the developmental processes of these organs [2,13]. This is considered to be closely related to the postnatal expression of DDO, which is the only enzyme involved in degradation of d-aspartic acid [36]. Functional maturation in rat salivary glands progresses over a period of 4 to 5 weeks after birth [65]. d-Aspartic acid levels were high in rat salivary glands at 3 to 7 weeks after birth [66]. The present finding showing that protein levels of DDO were lower (less than one tenth) in the three major salivary glands in 7-week-old rats than in the cerebral cortex or cerebellum is in good agreement with earlier findings [36,66].

An earlier study found high amounts of L-glutamic acid and low %D values in the following peripheral endocrine organs in rat: the thymus (12,533.7 ± 2931.7 nmol/g wet tissue, %D = 0.09%), pancreas (12,710.4 ± 3351.1 nmol/g wet tissue, %D = 0.01%) and adrenal gland (1424.2 ± 258.6 nmol/g, %D = 0.05%) [67]. In the present study, the amounts of l-glutamic acid detected in the three major salivary glands were lower than those found in these peripheral endocrine organs, except for where this was lowest, in the adrenal gland. Meanwhile, d-glutamic acid was only detected at trace levels in all three salivary glands. This discrepancy may be due to the different number of amino acid enantiomers analyzed. That is, the earlier study focused on only two enantiomers of acidic amino acids (glutamic acid and aspartic acid), which may explain the higher detection sensitivity than that in the present study.

Glutamate transporter-1 (GLT-1) and glutamate aspartate transporter (GLAST) can actively take up extracellular d/l-aspartic acid to cytoplasm [68,69]. One previous study demonstrated that ductal cells and acinar cells in the submandibular glands in rat showed high protein expression of GLT-1 and GLAST, respectively [70]. These findings suggest that these transporters play an important role in the absorption, disposition, and elimination of l-glutamic acid and d/l-aspartic acid in salivary gland.

One earlier study [71] demonstrated lower levels of d-aspartic acid (7.8 nmol/g) in rat salivary gland than observed in the present or another previous study [66]. This discrepancy may be explained by diet-related differences in time elapsed following consumption of food prior to sampling, which is supported by the following results: (1) in one of the previous studies [71] and the present study, the rats were allowed to feed on CE-2 (Clea Japan Inc., Tokyo, Japan), which is rich in d-aspartic acid (50 ± 1 nmol/g) [72]; (2) following consumption of 20 mM d-aspartic acid in drinking water for 12 days, d-aspartic acid levels increased by approximately two- and seven-fold in comparison with at the basal level in serum and testis, respectively, in rat [73]; (3) at 12 h after termination of administration of d-aspartic acid, the amount of d-aspartic acid in serum and testis decreased to equal to and 2-fold of the basal level, respectively [73]; (4) intravenous administration of ^14^C-labeled d-aspartic acid resulted in high radioactivity at 0.5 h in salivary gland in rat, which subsequently disappeared almost completely over the next 3 to 24 h [71].

Both d-serine and d-alanine acts as an endogenous coagonist for the glycine site on NR1 of the NMDA receptor. Earlier studies showed that the amount of d-alanine in the brain was much lower than that of d-serine, but that the amount of d-alanine in the peripheral tissues was higher than that of d-serine [26,34]. The results of the present study correspond well with these earlier findings.

Expression of serine racemase was observed in salivary glands, suggesting it may produce endogenous d-serine there. One recent study reported that serine racemase was expressed in islets and regulated insulin contents through NMDA receptors [74]. This suggests that d-serine produced by serine racemase plays a physiological role in peripheral organs as well as in the central nervous system.

The fact that DAO was expressed in rat salivary glands provides further support for the view that it metabolizes endogenous d-serine and d-alanine in salivary glands. Further support for this comes from the fact that the amounts of d-serine and d-alanine showed a significant increase in a range of peripheral tissues, including the pancreas, liver and kidney, in Long–Evans agouti/SENDAI (LEA/Sen) rats lacking DAO compared to in Wistar or SD rats [75]. It is, however, important to note that the substrate specificity and activity of rat DAO is different from that of human DAO, and that the catabolic efficiency of d-serine is low [76]. Further studies using rats lacking DAO are needed to clarify the contribution of DAO to the catabolism of neutral and basic d-amino acids, including d-serine and d-alanine.

Glycine plays a role in inhibitory and excitatory neurotransmission by binding the strychnine-sensitive glycine receptor and NMDA receptor, respectively. Glycine is catabolized to carbon dioxide and ammonium ions, or converted to l-serine in glial cells [77,78]. The results of the present study revealed that glycine levels in the salivary glands were substantially higher than those observed in cerebral cortex (676 nmol/g) or plasma (410 nmol/mL) [79].

Different NR2 subtypes mediate distinct physiological functions [80,81]. In the forebrain, most NMDA receptors are made up of four subunits (two NR1 paired with two NR2A, two NR2B, or one NR2A and one NR2B) [82], and triheteromeric NMDA receptors have been reported to mediate NMDA-induced toxicity [83]. The result of the present study demonstrated that salivary glands expressed NR2D, but not NR2A-C, a pattern that has also been reported in the retina [84]. In the retina, rod bipolar cells expressing NR2D are resistant to excitotoxicity by calcium influx [85,86]. Thus, these findings correspond well with an earlier finding showing high concentrations of normal extracellular concentration of Ca^2+^ in salivary gland (2.56 mM) [87] compared to in hippocampus (0.5 mM) [88].

Furthermore, salivary glands show normal extracellular concentration of 1.1 mM Mg^2+^ [87]. Basal levels of Mg^2+^ are tightly controlled and regulate enzyme secretion and several membrane ion transport systems in salivary glands [87,89]. Triheteromeric (NR1/2A/2B, NR1/2A/2C and NR1/2B/2D) and diheteromeric (NR1/2A and NR1/2B) NMDA receptors are more sensitive to voltage-dependent block of NMDA receptors induced by extracellular Mg^2+^ (~1 mM) than NR1/2D [90]. One earlier study reported that addition of 1.2 mM Mg^2+^ prevented CRF release from cultured rat amygdala neuron cells by L-glutamic acid [91]. These results indicate that the diheteromeric NR1/2D subtypes contribute to the activation of NMDA receptors in salivary glands, suggesting that they play a physiological role.

The levels of serine racemase, DAO, NR1, and NR2D protein expression in salivary glands and cerebral cortex differed from their mRNA expression levels. Several reports have demonstrated that expression of mRNA does not necessarily reflect a corresponding level of protein [92,93]. Further support for this comes from the fact that the protein levels of serine racemase and DAO in several brain areas did not agree with their gene expression [54].

In addition to NR2D, NR1 was also detected in salivary gland in the present study. An earlier immunohistochemical study reported that no immunoreactivity was detected for NR1 in acinar cells, ductal cells in submandibular glands, or submandibular ganglion; however, the data were not presented [94]. At least for gene and protein expression of NR1, the results of present study showed that RT-PCR and Simple Western assay were more sensitive and specific than immunohistochemistry. Indeed, earlier studies have also demonstrated that immunohistochemistry was less sensitive and specific than RT-PCR [92] or Simple Western [95]. However, further studies are needed to clarify localization of NMDA receptor in salivary glands.

## 5. Conclusions

A large amount of d-aspartic acid was detected together with small amounts of d-serine and d-alanine in all three major salivary glands in rat in the present study. In addition, NMDA receptor subunit, NR1 and NR2D, protein was detected in all three major salivary glands. Although further biochemical and physiological analyses are needed to evaluate the functions of these d-amino acids in vivo, the present results suggest that d-amino acids play a physiological role in a range of endocrine and exocrine function in salivary glands.

## Figures and Tables

**Figure 1 biology-11-00390-f001:**
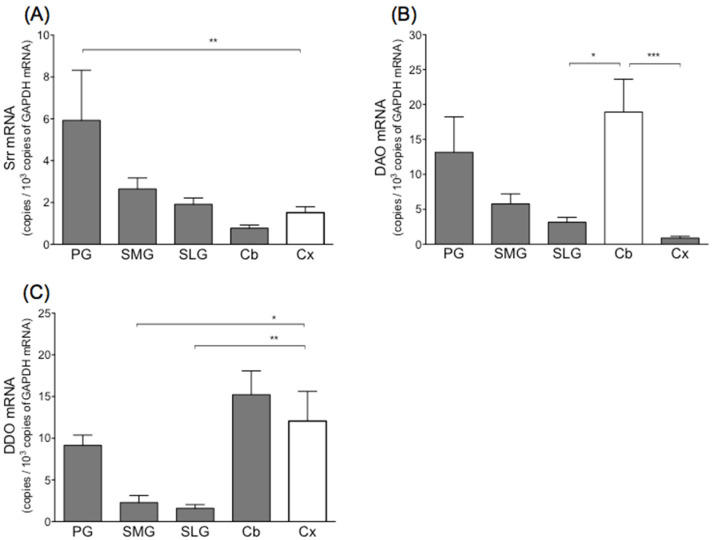
mRNA levels of serine racemase (Srr) (**A**), d-amino acid oxidase (DAO) (**B**) and d-aspartate oxidase (DDO) (**C**) in three salivary glands and two brain areas (cerebral cortex and cerebellum) in rat. Values represent mean ± SD in 6 rats. Significantly different from cerebral cortex (Srr, DDO) or cerebellum (DAO) according to Dunn’s post hoc test following Kruskal–Wallis test; * *p* < 0.05, ** *p* < 0.01, and *** *p* < 0.001. PG, parotid gland; SMG, submandibular gland; SLG, sublingual gland; Cb, cerebellum; Cx, cerebral cortex.

**Figure 2 biology-11-00390-f002:**
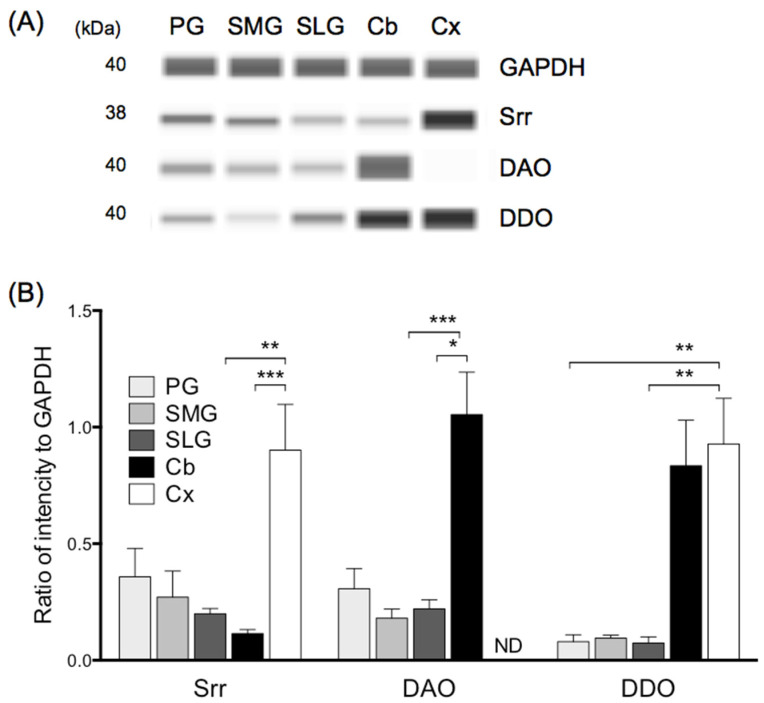
Protein levels of serine racemase (Srr), d-amino acid oxidase (DAO) and d-aspartate oxidase (DDO) in three salivary glands, cerebellum, and cerebral cortex in rat. (**A**) Typical Simple Western image of Srr, DAO, DDO and glyceraldehyde-3-phosphate dehydrogenase (GAPDH). (**B**) Values were normalized to housekeeping gene GAPDH. Values represent mean ± SD in 5 rats. Significantly different from cerebral cortex (Srr, DDO) or cerebellum (DAO) according to Dunn’s post hoc test following Kruskal–Wallis test; * *p* < 0.05, ** *p* < 0.01, and *** *p* < 0.001. PG, parotid gland; SMG, submandibular gland; SLG, sublingual gland; Cb, cerebellum; Cx, cerebral cortex. ND; not detected.

**Figure 3 biology-11-00390-f003:**
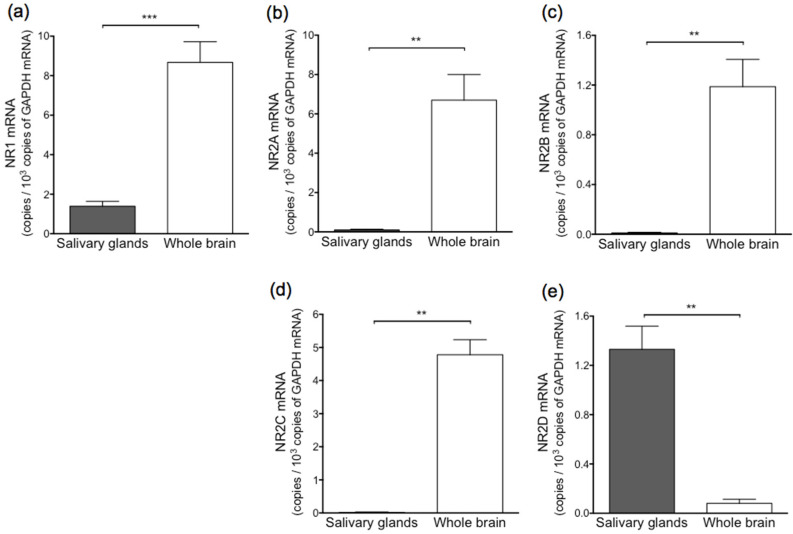
mRNA levels of NR1 (**a**), NR2A (**b**), NR2B (**c**), NR2C (**d**), and NR2D (**e**) in mixture of three salivary glands and whole brain of rat. Values represent mean ± SD in 5 rats. Significantly different according to Mann–Whitney test between two groups; ** *p* < 0.01, and *** *p* < 0.001.

**Figure 4 biology-11-00390-f004:**
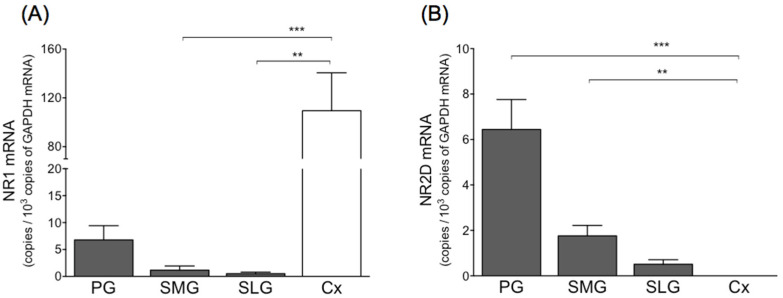
mRNA levels of NR1 (**A**) and NR2D (**B**) in three salivary glands and cerebral cortex in rat. Values represent mean ± SD in 6 rats. Significantly different from cerebral cortex according to Dunn’s post hoc test following Kruskal–Wallis test; ** *p* < 0.01, and *** *p* < 0.001. PG, parotid gland; SMG, submandibular gland; SLG, sublingual gland; Cx, cerebral cortex.

**Figure 5 biology-11-00390-f005:**
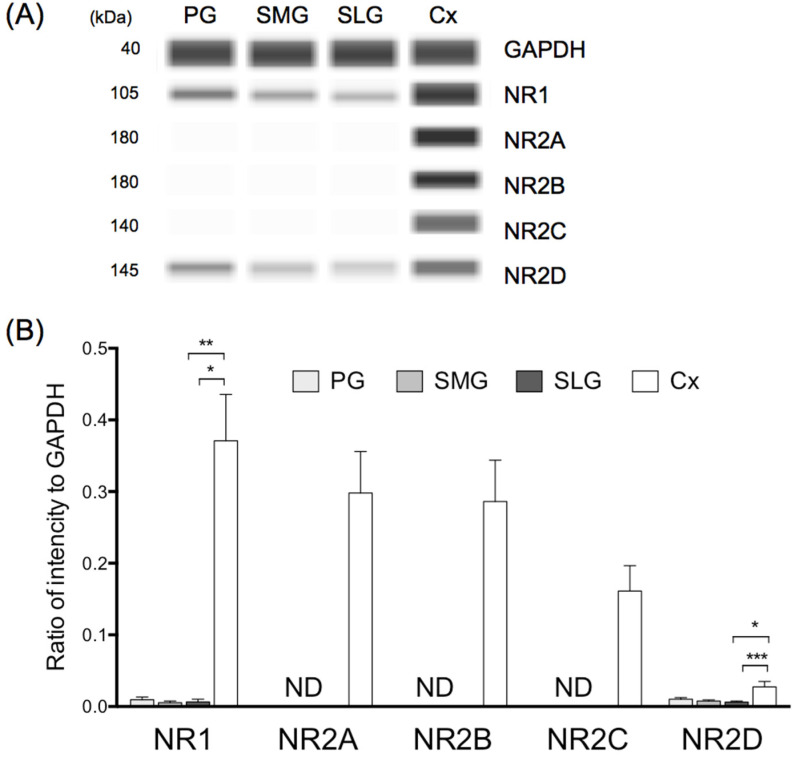
Protein levels of NR1 and NR2A-D in three salivary glands and cerebral cortex in rat. (**A**) Typical Simple Western image of NR1, NR2A-D and glyceraldehyde-3-phosphate dehydrogenase (GAPDH). Values represent mean ± SD in 5 rats. (**B**) Values were normalized to housekeeping genes GAPDH. Significantly different from cerebral cortex according to Dunn’s post hoc test following Kruskal–Wallis test; * *p* < 0.05, ** *p* < 0.01, and *** *p* < 0.001. PG, parotid gland; SMG, submandibular gland; SLG, sublingual gland; Cx, cerebral cortex. ND; not detected.

**Table 1 biology-11-00390-t001:** Amino acids contents in three major salivary glands of rats.

	Parotid Gland			Submandibular Gland			Sublingual Gland		
	D (nmol/g)	L (nmol/g)	D/(D + L) (%)	D (nmol/g)	L (nmol/g)	D/(D + L) (%)	D (nmol/g)	L (nmol/g)	D/(D + L) (%)
His	N.D.	406.2 ± 76.3	–	N.D.	751.8 ± 110.8	–	N.D.	582.3 ± 86.3	–
Asn	N.D.	480.0 ± 79.2	–	N.D.	998.0 ± 137.4	–	N.D.	997.6 ± 47.4	–
Ser	3.8 ± 0.5	1841.3 ± 199.4	0.2 ± 0.05	4.9 ± 0.5	3530.4 ± 246.2	0.1 ± 0.02	4.3 ± 0.3	3116.1 ± 164.6	0.1 ± 0.01
Gln	N.D.	1959.4 ± 221.0	–	N.D.	3603.8 ± 368.0	–	N.D.	3068.8 ± 320.9	–
Arg	trace	309.6 ± 99.1	–	trace	4547.1 ± 1321.9	–	trace	2678.5 ± 689.1	–
Asp	143.4 ± 65.7	786.0 ± 97.1	15.5 ± 7.4	174.3 ± 53.8	1381.9 ± 163.9	11.1 ± 3.1	104.7 ± 24.8	1833.4 ± 134.3	5.3 ± 1.0
Gly	–	3538.3 ± 614.8	–	–	4483.4 ± 440.2	–	–	4450.3 ± 243.9	–
allo-thr	N.D.	N.D.	–	N.D.	N.D.	–	N.D.	N.D.	–
Glu	trace	3670.2 ± 431.2	–	trace	3933.5 ± 319.9	–	trace	4625.5 ± 379.2	–
Thr	N.D.	837.9 ± 64.1	–	N.D.	1644.3 ± 171.31	–	N.D.	1564.0 ± 121.91	–
Ala	11.6 ± 8.8	3285.3 ± 461.3	0.3 ± 0.2	14.1 ± 8.1	6405.0 ± 531.9	0.2 ± 0.1	13.0 ± 7.9	5680.3 ± 151.5	0.2 ± 0.1
Pro	N.D.	1131.4 ± 44.1	–	N.D.	2641.7 ± 308.0	–	N.D.	2080.6 ± 178.5	–
Met	N.D.	262.8 ± 19.8	–	N.D.	742.5 ± 118.4	–	N.D.	644.5 ± 75.6	–
Val	N.D.	906 ± 180.7	–	N.D.	1845.6 ± 316.5	–	N.D.	1644.0 ± 206.6	–
allo-Ile	N.D.	N.D.	–	N.D.	N.D.	–	N.D.	N.D.	–
Ile	N.D.	363.1 ± 15.3	–	N.D.	1641.5 ± 361.5	–	N.D.	1266.6 ± 249.0	–
Leu	N.D.	1686.4 ± 303.8	–	N.D.	4093.9 ± 494.8	–	N.D.	3037.6 ± 376.20	–
Phe	N.D.	791.4 ± 142.2	–	N.D.	2205.2 ± 542.4	–	N.D.	1600.6 ± 355.3	–
Trp	N.D.	236.5 ± 40.6	–	N.D.	261.1 ± 40.4	–	N.D.	222.9 ± 11.4	–
Lys	N.D.	1153.9 ± 92.1	–	N.D.	2197.2 ± 413.0	–	N.D.	2263.1 ± 231.4	–
Cystein	N.D.	84.9 ± 70.1	–	N.D.	17.2 ± 11.2	–	N.D.	96.5 ± 26.4	–
Tyr	N.D.	303.0 ± 64.7	–	N.D.	877.9 ± 449.9	–	N.D.	3597.1 ± 514.6	–

The results are expressed as the mean ± SD from 3 rats. “N.D.” = not detected.

## Data Availability

The data that support the findings of this study are available from the corresponding author, [M.Y.], upon reasonable request.

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
