# Peer review of "Free d-Amino Acids in Salivary Gland in Rat"

_biology, 2022, doi:10.3390/biology11030390_

Round 1

Reviewer 1 Report

The authors answered many of the concerns arose by this reviewer

Author Response

We thank the reviewer for his/her careful reading our manuscript and for your helpful suggestions.

Reviewer 2 Report

The paper by Yoshikawa et al. reports about a detailed analysis of D-amino acid content together with analysis of mRNA and protein levels of the three enzymes involved in D-amino acid metabolism in mammals in rat salivary glands. The topic is surely interesting in view of the increasing findings about D-amino acids and their role in humans, especially since it evidences the presence of the main three D-amino acids found in mammals in a tissue up to now not investigated about this aspect. Methods are adequately described and results are clearly reported even with the help of appropriate figures and discussed. Discussion and conclusions are fully supported by the results obtained. The paper can be accepted in present form after correction of minor mistakes:

Line 13: “D-alanine” should be written instead of “L-alanine”

Lines 55 and 83: “mammals” should be written instead of “mammal”

Author Response

The paper by Yoshikawa et al. reports about a detailed analysis of D-amino acid content together with analysis of mRNA and protein levels of the three enzymes involved in D-amino acid metabolism in mammals in rat salivary glands. The topic is surely interesting in view of the increasing findings about D-amino acids and their role in humans, especially since it evidences the presence of the main three D-amino acids found in mammals in a tissue up to now not investigated about this aspect. Methods are adequately described and results are clearly reported even with the help of appropriate figures and discussed. Discussion and conclusions are fully supported by the results obtained. The paper can be accepted in present form after correction of minor mistakes:

Response: We thank the reviewer for his/her kind remark.

Line 13: “D-alanine” should be written instead of “L-alanine”

Response: We thank the reviewer for raising this point, and correcting us. We have changed it. Please see line 13.

Lines 55 and 83: “mammals” should be written instead of “mammal”

Response: We thank the reviewer for raising this point, and correcting us. We have changed it. Please see line 59 and 88.

Reviewer 3 Report

Dear Authors, 

In the paper, the authors report that in parotid, submandibular, and sublingual glands of rats they found high concentrations of D-aspartic acid and low ones of D-serine and L-alanine. Besides these, these glands also contain serine racemase,  as was N-methyl- D-aspartic acid receptor subunits NR1 and NR2D, but not NR2A, NR2B, or NR2C. 

The paper covers a relevant topic. The study is well-conducted and designed. 

  • Abstract: The abstract lacks a summary and explanation of the results. Please summarize this in one sentence.
  • Figure 3 The markings on the sub-figures are a bit confusing. It would be better to mark each sub-figure here with A, B, C, etc., as they did in the other figures.
  • Formal errors: space error occurs in several places, for example:

Line 55 double space before Maximal

Line 81 double space before [36]

Line 92 missing space expression[41, 42]

Line 167 missing space (GAPDH)(GenBank

Line 170 missing space mRNA(accession

Please check and correct them.

Overall, I found the report quite innovative, very interesting and scientifically sound. 

I recommend this manuscript for publication after minor revision.

Author Response

In the paper, the authors report that in parotid, submandibular, and sublingual glands of rats they found high concentrations of D-aspartic acid and low ones of D-serine and L-alanine. Besides these, these glands also contain serine racemase,  as was N-methyl- D-aspartic acid receptor subunits NR1 and NR2D, but not NR2A, NR2B, or NR2C. 

The paper covers a relevant topic. The study is well-conducted and designed. 

Response: We thank the reviewer for his/her kind remark.

Abstract: The abstract lacks a summary and explanation of the results. Please summarize this in one sentence.

Response: We thank the reviewer for raising this point. We have added sentence as follows; “The results of the present study suggest that D-amino acids play a physiological role in a range of endocrine and exocrine function in salivary glands.” Please see lines 28-29.

Figure 3 The markings on the sub-figures are a bit confusing. It would be better to mark each sub-figure here with A, B, C, etc., as they did in the other figures.

Response: We thank the reviewer for raising this good point. Based on the reviewer’s comments, we marked each sub-figure with I, ii, iii, …, Please see figure 3 and its legends.

Formal errors: space error occurs in several places, for example:

Response: We thank the reviewer for raising this point, and correcting us.

Line 55 double space before Maximal

Response: We removed one space.

Line 81 double space before [36]

Response: We removed one space.

Line 92 missing space expression[41, 42]

Response: We added one space.

Line 167 missing space (GAPDH)(GenBank

Response: We added one space.

Line 170 missing space mRNA(accession

Response: We added one space.

Overall, I found the report quite innovative, very interesting and scientifically sound. 

I recommend this manuscript for publication after minor revision.

Response: We thank the reviewer for his/her careful reading our manuscript and for your helpful suggestions.

This manuscript is a resubmission of an earlier submission. The following is a list of the peer review reports and author responses from that submission.

Round 1

Reviewer 1 Report

The paper from Yoshikawa et al. focuses on the identification of the concentration of D-amino acids in salivary glands, saliva, plasma and interstitial fluids of mice. Indeed, the transcriptional levels of SR, DAO and NMDAR subunits in salivary glands and selected brain regions were also determined. The paper reports quite novel results, although in some cases the work seems too preliminary. Further analyses, controls and a text revision will significantly improve the study.

Some points, from my side, that need a comment/revision from the authors:

  • the authors did not use an enzymatic treatment to eliminate D-amino acids before HPLC analysis, as suggested in the guidelines from Mothet et al., 2019. Could they state clearly the controls performed to validate the determined D-amino acid levels?
  • Lines 55-56: the presence of an Aspartate Racemase activity in mammals is still debate: cite here the related works.
  • Table 1: Cystin or Cysteine?
  • Paragraph 3.2 and related figures: these seem a set up of the experimental conditions for microdialysis determinations. This seems most suited for Materials and Methods section since no results are provided.
  • Figure 4: how do you explain the 120 min required to reach a stable AA level?
  • Transcriptomic analyses: here, the detection of protein levels of SR, DAO, D-aspartate oxidase and NMDAR subunits is absolutely required to explain the D-amino acids levels.
  • The Discussion section seems a review taking into consideration different points. I suggest to revise it, by shortening the text and focusing on the results gathered in the paper and the related meaning. E.g., text at lines 357-367 can be deleted, as well as at lines 393-398.
  • Lines 371-372: you cannot state this since the levels (protein and activity) of DAO were not determined (see above).
  • Lines 392-393: what is the meaning of this sentence? What “methodological differences” means?
  • Lines 419-424: the presence and level of these bacteria have not been evaluated. This conclusion is with no experimental evidence. This part should be eliminated and a different conclusion should be provided.
  • I’ve a strong criticism about the selected references cited by the authors. I know the huge contribution given by Japanese researchers to the field, but a more equilibrated selection of cited papers is required. I mean: a) among the starting 10 citations only one was not from Japanese groups; b) Refs. 20, 21: I feel the work of other researchers can be cited as well; c) lines 334-343: the work of D’Aniello’s group should be cited; d) among Refs. 64-66 the review from Sweedler’s lab should be cited too; etc…...

Author Response

The paper from Yoshikawa et al. focuses on the identification of the concentration of D-amino acids in salivary glands, saliva, plasma and interstitial fluids of mice. Indeed, the transcriptional levels of SR, DAO and NMDAR subunits in salivary glands and selected brain regions were also determined. The paper reports quite novel results, although in some cases the work seems too preliminary. Further analyses, controls and a text revision will significantly improve the study.

Response: We thank the reviewer for his/her kind remark.

Some points, from my side, that need a comment/revision from the authors:the authors did not use an enzymatic treatment to eliminate D‑amino acids before HPLC analysis, as suggested in the guidelines from Mothet et al., 2019. Could they state clearly the controls performed to validate the determined D‑amino acid levels?

Response: We think that the reviewer makes a reasonable request. The greatest advantage of microdialysis is that it allows quantitative changes in D -amino acids to be measured over time. The sample volume adopted in the present study was 50 microliters. We analyzed 50-microliters samples of dialysate by HPLC after first treating them with DAO. However, it was impossible to sufficiently separate the DAO-derived contaminants in the sample solution. We tried several pretreatments aimed at removing these contaminants. As a sham test, samples that had not been treated with DAO were pretreated to remove contaminants. The amino acids were below the detection limit allowed by the HPLC analysis, however. It was also impossible to analyze the amino acids after enzymatic treatment of the dialysate. Thus, enzymatic treatment could not be performed in accordance with the guidelines of Mothet et al. It would be impossible to verify the determined D-amino acid levels as the reviewers suggest. Therefore, all data on microdialysis have been removed from the revised manuscript. We have changed title. Further studies using rats lacking DAO is needed to validate the determined D-amino acid levels.

Lines 55-56: the presence of an Aspartate Racemase activity in mammals is still debate: cite here the related works

Response: We thank the reviewer for their comment. We have addressed these points in abstract as requested. Please see lines 61: “On the other hand, no aspartate racemase has been identified in mammals so far, although serine racemase is suggested to act as a biosynthetic enzyme of D-aspartic acid in some tissue [29].”

Table 1: Cystin or Cysteine?

Response: We thank the reviewer for raising this point, and correcting us. We analyzed Cysteine in salivary glands. Please see Table 1.

Paragraph 3.2 and related figures: these seem a set up of the experimental conditions for microdialysis determinations. This seems most suited for Materials and Methods section since no results are provided.

Response: We thank the reviewer for this suggestion. As mentioned above, all data on microdialysis have been removed from the revised manuscript.

Figure 4: how do you explain the 120 min required to reach a stable AA level?

Response: We thank the reviewer for raising this good point. We judged the recovery rate as stable when the average of the quantitative values of each amino acid in the three consecutive fractions was within 10% of the difference between those of the three fractions before and after.

Transcriptomic analyses: here, the detection of protein levels of SR, DAO, D-aspartate oxidase and NMDAR subunits is absolutely required to explain the D-amino acids levels.

Response: We thank the reviewer for raising this good point. Based on the reviewer’s comments, we added the study of capillary electrophoresis‑based immunodetection assay (Simple Western) for determination of protein expression levels of serine racemase, D‑amino acid oxidase (DAO), D-aspartate oxidase (DDO), and NMDA receptor subunits (Figure 2 and 5). These assays revealed following things. First, “protein expression of serine racemase was observed in salivary glands, suggesting it may produce endogenous D-serine there”(lines 434-435). Second, “The fact that DAO was expressed in rat salivary glands, provides further support for the view that it metabolizes endogenous D-serine and D-alanine in salivary glands” (lines 440-442). Third, “protein levels of DDO were lower (less than one tenth) in the three major salivary glands in 7‑week‑old rats than in the cerebral cortex and cerebellum is in good agreement with the these earlier findings” (lines 388-391). Fourth, NMDA receptor subunit proteins NR1 and NR2D were detected in all three major salivary glands (Figure 5). Fifth, “The levels of serine racemase, DAO, NR1, and NR2D protein expression in salivary glands and cerebral cortex differed from their mRNA expression levels” (lines 483-485).

The Discussion section seems a review taking into consideration different points. I suggest to revise it, by shortening the text and focusing on the results gathered in the paper and the related meaning. E.g., text at lines 357-367 can be deleted, as well as at lines 393-398.

Response: We thank the reviewer for raising this point. We have addressed these points in discussion as requested. Please see discussion section.

Lines 371-372: you cannot state this since the levels (protein and activity) of DAO were not determined (see above).

Response: Based on the reviewer’s comments, we determined protein expression levels. Please see above.

Lines 392-393: what is the meaning of this sentence? What “methodological differences” means?

Response: We apologize for not being clear about this in the original submission. We have changed this sentence as follows; “At least for gene and protein expression of NR1, the results of present study showed that RT‑PCR and Simple Western assay were more sensitive and specific than immunohistochemistry.” Please see lines 494-496.

Lines 419-424: the presence and level of these bacteria have not been evaluated. This conclusion is with no experimental evidence. This part should be eliminated and a different conclusion should be provided.

Response: We thank the reviewer for raising this point, and correcting us. We have deleted this statement.

I’ve a strong criticism about the selected references cited by the authors. I know the huge contribution given by Japanese researchers to the field, but a more equilibrated selection of cited papers is required. I mean: a) among the starting 10 citations only one was not from Japanese groups; b) Refs. 20, 21: I feel the work of other researchers can be cited as well; c) lines 334-343: the work of D’Aniello’s group should be cited; d) among Refs. 64-66 the review from Sweedler’s lab should be cited too; etc…...

Response: We thank the reviewer for raising this good point. Based on the reviewer’s comments, we have addressed these points. 

Reviewer 2 Report

This report revealed the localization and kinetics of D-amino acids in salivary glands and the expression level of SRR and DAO mRNA. It seems to be useful for future functional analysis and biomarker search as basic data of D-amino acids in salivary glands.  

There are a few corrections shown below. 

line 148; Plasma separation is usually performed at 4 ° C to ensure amino acid stability, but make sure it is correct at room temperature. 

Result 3.5; D-Aspartate oxidase has also been reported to be present in mammals, but why not measure it this study? DAO is also an enzyme that also metabolizes D-Ala, but please explain in an easy-to-understand why you focused only on the metabolism of D-Ser this time. 

line 428; Isn't the submandibular glands correctly salivary glands? 

Overall; The D- and L-fonts should be about 2 points smaller .

Author Response

This report revealed the localization and kinetics of D‑amino acids in salivary glands and the expression level of SRR and DAO mRNA. It seems to be useful for future functional analysis and biomarker search as basic data of D‑amino acids in salivary glands.

Response: We thank the reviewer for his/her kind remark.

There are a few corrections shown below.

line 148; Plasma separation is usually performed at 4°C to ensure amino acid stability, but make sure it is correct at room temperature.

Response: We thank the reviewer for raising this good point, and correcting us. We separated plasma at 4°C. Base on other reviewers’ comments, all data on microdialysis and physiological fluid have been removed from the revised manuscript, however. We have changed title.

Result 3.5; D-Aspartate oxidase has also been reported to be present in mammals, but why not measure it this study? DAO is also an enzyme that also metabolizes D-Ala, but please explain in an easy-to-understand why you focused only on the metabolism of D-Ser this time.

Response: We thank the reviewer for raising this good point. We determined protein expression levels of serine racemase, D‑amino acid oxidase (DAO), D‑aspartate oxidase (DDO), and NMDA receptor subunits (Figure 2 and 5), in addition to gene expression level of DDO in salivary glands (Figure 1C). These assays revealed following things. First, “Expression of serine racemase was observed in salivary glands, suggesting it may produce endogenous D-serine there”(lines 432-434). Second, “The fact that DAO was expressed in rat salivary glands, provides further support for the view that it metabolizes endogenous D-serine and D-alanine in salivary glands” (lines 440-442). Third, “protein levels of DDO were lower (less than one tenth) in the three major salivary glands in 7‑week‑old rats than in the cerebral cortex and cerebellum is in good agreement with the these earlier findings” (lines 388-391). Fourth, NMDA receptor subunit proteins NR1 and NR2D were detected in all three major salivary glands (Figure 5). Fifth, “The levels of serine racemase, DAO, NR1, and NR2D protein expression in salivary glands and cerebral cortex differed from their mRNA expression levels” (lines 483-485). Sixth, “Both D‑serine and D‑alanine acts as an endogenous coagonist for the glycine site on NR1 of the NMDA receptor. Earlier studies showed that the amount of D‑alanine in the brain was much lower than that of D‑serine, but that the amount of D‑alanine in the peripheral tissues was higher than that of D‑serine. The results of the present study correspond well with these earlier findings.” (lines 428-433).

line 428; Isn't the submandibular glands correctly salivary glands?

Response: We thank the reviewer for raising this point, and correcting us. We have now corrected the text. Please see conclusion section.

Overall; The D‑ and L‑fonts should be about 2 points smaller .

Response: We thank the reviewer for raising this point, and correcting us. We have corrected the font size in the text.

Reviewer 3 Report

In this manuscript Yoshikawa and coworkers analyzed the cellular and extracellular content of free D-amino acids (D-serine, D-aspartate and D-alanine) and in the three major salivary glands in rat. They also investigated the expression of D-serine metabolic enzymes (serine racemase and D-amino acid oxidase) and NMDA receptors (different subunits) by determining their transcripts levels in the same tissues.

The strength of the paper lies in the novelty of the reported results, rather than in the methodologies used: as stated by the authors themselves this is the first study investigating D-amino acids levels in the interstitial fluid of peripheral tissues. Unfortunately however, I think this manuscript will mainly catch the attention of the “insiders”.

The weaknesses of the submitted manuscript in my opinion are: the incompleteness of the performed analyses (D-aspartate oxidase expression, as well as glutamate and glycine levels in the interstitial fluids are missing) that unfortunately makes the proposed conclusions not very consistent. It’s true that the study is presented as preliminary in the simple summary, but the authors should have highlighted them in the discussion paragraph.

Also, although the proposed function of D-aspartate and D-serine is proposed (and not demonstrated), the other aim that the authors have set for their manuscript (namely establish the potential of D-amino acids as salivary biomarkers) appears at least smoky. With respect to what free D-amino acids may represent valuable biomarker remain to be detailed and explained.

For these reasons I suggested a revision of the manuscript before resubmission for publication.

Minor revisions:

Line 38 – Please change reference as follow [1, 5, 6].

Line 40 – Reference needed. Suggestion: Mothet et al. (2000) PNAS; 97:4926

Line 53 – “D-serine and D-aspartate, is tissue; that is…”. As a suggestion, this should be changed in: “D-serine and D-aspartate, is tissue: they are synthesized (or produced) by (or through)….”

Line 56, 57 – “Fermented foods are enriched… “. A reference is needed. Suggestion: Marcone et al. (2020) Appl Microbiol Biotechnol; 104:555

Line 61 – Please change “D-amino acid oxidase” in “DAO”. Please change “(EC 1.4.3.15)” in “(DDO, EC 1.4.3.15)”.

Line 69-71 – based on the authors’ considerations, here is not clear for what (or in which context) they propose free D-amino acids as salivary biomarkers

Line 72 – “The localization od D-amino acids…“ please, add another reference better supporting this statement. Ref 25 is an editorial presenting a special issue.

Line 75 – Recent findings indicated that actually D-serine can’t be defined as a neurotransmitter. Better to refer to this molecule as a neuromodulator. For a recent review see Coyle et al. (2020) Nerochem Research; 4:1344

Line 81 – Could the authors explain why in the interstitial fluid glycine content (as well as glutamate one) was not determined?

Line 86 – Analogously, why the authors did not analyze DDO gene expression?

Line 161, 162 – Please change “ten” and “fourty” in “10” and “40”, respectively.

Results Fig. 3 – Differently from serine and alanine, here the concentration of D-aspartate enantiomers analyzed as standards is the same. Why?

Major points:

Results 3.1 – I think the author should also discuss L-enantiomers levels and/or (at least) about the D/(D+L)% ratio, which have been determined and reported in table 1. Furthermore, glycine and glutamate deserve a little attention too, in my opinion.

Results 3.3 – It would have been better to also analyze glutamate and glycine content in the dyalisate (interstitial fluid). The results would have important to complete the picture and perhaps would have allowed to formulate further conclusions.

Discussion, line 329-331 – Concerning D-aspartate levels, I think it is a pity not to have determined DDO expression levels in the three major salivary glands. I would have been possible to compare them to DAOP ones and would have probably helped explain the elevated levels of the acidic D-amino acid in these tissues.

Discussion, line 341-343 – Here, further analyses at different ontological stages will strengthen this hypothesis. The authors should mention it.

Discussion, line 344-356 – So, in my understanding the principal source of D-aspartate in these animals is from diet. I think the authors should discuss how this fit with what they proposed above (line 341-343).

Discussion– Concerning the role of DAO in controlling D-serine levels in salivary glands, I think the authors should consider and discuss the extremely low activity reported for rat DAO on D-serine and Dalanine as substrates. Please see Frattini et al. (2011) FEBS J; 278:4362.

Discussion, line 370-371 – The authors reported DAO (as well as serine racemase) encoding gene transcription levels in salivary glands. In my opinion DAO expression (by Western blot analyses), and eventually activity, should be investigated to confirm the suggested role in regulating D-serine levels. The authors should discuss about this.

Discussion, line 380-388 – Here, measurements of glutamate and glycine concentrations would have been extremely important to substantiate the proposed notions. Without these data, I’m afraid that the reported observations are poorly consistent.

Discussion – The authors should mention that D-alanine is a potent co-agonist of NMDA receptors. In the brain, its levels are very low (much lower that D-serine ones), but here the reported levels in the interstitial fluid are even higher compared to D-serine.

Discussion – Expression of NMDA receptor subunit NR2D is impressive. Could this subunit be regarded as peculiar for salivary glands? How the presence of this subunit in the tetrameric NR1/NR2D receptors may affect glutamate binding affinity? The authors think that NMDA receptors are involved in ionotropic or metabotropic signaling (or in both) in salivary glands?

Author Response

In this manuscript Yoshikawa and coworkers analyzed the cellular and extracellular content of free D‑amino acids (D‑serine, D‑aspartate and D‑alanine) and in the three major salivary glands in rat. They also investigated the expression of D‑serine metabolic enzymes (serine racemase and D‑amino acid oxidase) and NMDA receptors (different subunits) by determining their transcripts levels in the same tissues.

The strength of the paper lies in the novelty of the reported results, rather than in the methodologies used: as stated by the authors themselves this is the first study investigating D‑amino acids levels in the interstitial fluid of peripheral tissues. Unfortunately however, I think this manuscript will mainly catch the attention of the “insiders”.

Response: We thank the reviewer for his/her remark.

The weaknesses of the submitted manuscript in my opinion are: the incompleteness of the performed analyses (D‑aspartate oxidase expression, as well as glutamate and glycine levels in the interstitial fluids are missing) that unfortunately makes the proposed conclusions not very consistent. It’s true that the study is presented as preliminary in the simple summary, but the authors should have highlighted them in the discussion paragraph.

Also, although the proposed function of D‑aspartate and D‑serine is proposed (and not demonstrated), the other aim that the authors have set for their manuscript (namely establish the potential of D‑amino acids as salivary biomarkers) appears at least smoky. With respect to what free D‑amino acids may represent valuable biomarker remain to be detailed and explained.

Response: We think that the reviewer makes a reasonable request. In our revised manuscript, we focused on relationship between the contents of D‑amino acids and expression of their metabolic or catabolic enzymes in salivary glands. Please see discussion. We have followed the reviewer’s suggestion and now show gene and protein expression levels of D‑aspartate oxidase (DDO) in salivary glands. Please see Figure 1 and 2C. Base on other reviewer’s comments, all data on microdialysis and physiological fluid have been removed from the revised manuscript.

Minor revisions:

Line 38 – Please change reference as follow [1, 5, 6].

Response: We thank the reviewer for raising this point. We have changed this in abstract as follows; “D-Serine is known to be widely present in central nervous tissues, peripheral tissues, and body fluids [2, 6, 7].” Please see line 35.

Line 40 – Reference needed. Suggestion: Mothet et al. (2000) PNAS; 97:4926

Response: We appreciate the reviewer’s suggestion and have now included the reference in the text as follows; “D-Serine, in particular, is abundant in the mammalian forebrain, where it is involved in higher brain function during the entire postnatal period, acting as an endogenous and obligatory coagonist at the glycine site of the N-methyl-D-aspartic acid (NMDA) receptor [8, 9].” Please see lines 36-40.

Line 53 – “D‑serine and D‑aspartate, is tissue; that is…” As a suggestion, this should be changed in: “D‑serine and D‑aspartate, is tissue: they are synthesized (or produced) by (or through)….”

Response: We appreciate the reviewer’s suggestion and have addressed these points in abstract as requested as follows; “The first, which seems to be restricted to D-serine and D-aspartic acid, is tissue; these are synthesized by the racemization of their corresponding L‑amino acids.” Please see lines 58-60.

Line 56, 57 – “Fermented foods are enriched… “. A reference is needed. Suggestion: Marcone et al. (2020) Appl Microbiol Biotechnol; 104:555

Response: We appreciate the reviewer’s suggestion and have now added the reference as follows, “The second is diet. Fermented foods are enriched in D‑amino acids, including D‑alanine, D‑aspartic acid, D‑glutamic acid, and D‑proline [30].” Please see lines 64-65.

Line 61 – Please change “D‑amino acid oxidase” in “DAO”. Please change “(EC 1.4.3.15)” in “(DDO, EC 1.4.3.15)”.

Response: We thank the reviewer for raising this point. We have changed as follows, “D‑Serine and D‑aspartic acid are catabolized by D‑amino acid oxidase (DAO, EC 1.4.3.3) and D‑aspartate oxidase (DDO, EC 1.4.3.1), respectively, in mammals.” Please see lines 69-71.

Line 69-71 – based on the authors’ considerations, here is not clear for what (or in which context) they propose free D‑amino acids as salivary biomarkers

Response: We thank the reviewer for raising this point. We apologize because our studies do not show that using disease model animals led to changes in specific D‑amino acid content in saliva. We have now focus the relationship between the contents of D‑amino acids and expression of their metabolic or catabolic enzymes in salivary glands. To address the reviewer’s concerns in the revised text, we have done two things. 1) We have deleted “The aim of the present preliminary study was to elucidate the 15 function of D‑amino acids in salivary gland and establish their potential as salivary biomarkers.” in abstract (lines 14-15 in original submission) and stated “Several D‑amino acids have been observed in saliva, but their origin and the enzymes involved in their metabolism and catabolism remain to be clarified. In the present study, large amounts of D‑aspartic acid and small amounts of D‑serine and D‑alanine were detected in all three major salivary glands in rat.” in abstract (lines 18-21 in revised manuscript). 2) We have deleted “The aims of the present preliminary study were to elucidate the function of D‑amino acid in the salivary glands and establish their potential as salivary biomarkers.” in introduction (lines 81-82 in original submission). Please see last paragraph in introduction in revised manuscript (lines 96-103).

Line 72 – “The localization of D‑amino acids…“ please, add another reference better supporting this statement. Ref 25 is an editorial presenting a special issue.

Response: We thank the reviewer for raising this point. This paragraph including the sentence “The localization of D‑amino acids…”describes how the microdialysis is useful in revealing the localization of biological substances in the interstitial fulids.

Two other reviewers requested that we should use an enzymatic treatment to eliminate D-amino acids before HPLC analysis. We think that the reviewers make a reasonable request. The greatest advantage of microdialysis is that it allows quantitative changes in D-amino acids to be measured over time. The sample volume adopted in the present study was 50 microliters. We analyzed 50-microliters samples of dialysate by HPLC after first treating them with DAO. However, it was impossible to sufficiently separate the DAO-derived contaminants in the sample solution. We tried several pretreatments aimed at removing these contaminants. As a sham test, samples that had not been treated with DAO were pretreated to remove contaminants. The amino acids were below the detection limit allowed by the HPLC analysis, however. It was also impossible to analyze the amino acids after enzymatic treatment of the dialysate. Thus, enzymatic treatment could not be performed in accordance with the guidelines of Mothet et al. (2019). It would be impossible to verify the determined D-amino acid levels as the reviewers suggest. Unfortunately, all data on microdialysis have been removed from the revised manuscript. Moreover, we have changed title.

Line 75 – Recent findings indicated that actually D‑serine can’t be defined as a neurotransmitter. Better to refer to this molecule as a neuromodulator. For a recent review see Coyle et al. (2020) Nerochem Research; 4:1344.

Response: We thank the reviewer for raising this point. We have changed the sentences as follows; “Substantial amounts of D-serine were detected in extracellular space in the forebrain, where NMDA receptors are abundant [10], indicating that it is involved in glutamatergic neurotransmission via these receptors [11, 12].” Please see lines 40-42.

Line 81 – Could the authors explain why in the interstitial fluid glycine content (as well as glutamate one) was not determined?

Response: We apologize for not being clear about this important matter in the original submission. The HPLC system that used for determination of D‑ and L‑enantiomers in the interstitial fluid could not separate glycine and glutamate. As mentioned above, we have unfortunately removed all data on microdialysis from the revised manuscript according the other reviewers’ direction. In further study, we will analyze glycine and glutamate in the interstitial fluid using other HPLC methods.

Line 86 – Analogously, why the authors did not analyze DDO gene expression?

Response: We have followed the reviewer’s suggestion and now show gene and protein expression levels of DDO in salivary glands. Please see Figure 1 and 2C.

Line 161, 162 – Please change “ten” and “fourty” in “10” and “40”, respectively.

Response: We thank the reviewer for raising this point. As mentioned above, all data and methods on microdialysis have been removed from the revised manuscript. Therefore, the sentence you pointed out is not included in the revised manuscript.

Results Fig. 3 – Differently from serine and alanine, here the concentration of D-aspartate enantiomers analyzed as standards is the same. Why?

Response: We apologize for not being sufficient about this important matter in the original submission. In the prior study, D‑aspartic acid could not be detected in the dialysis sample, and L-aspartic acid was lower than that of other D‑amino acids. In order to investigate the conditions for higher sensitivity, we examined the conditions at a low concentration (1µM) for both D‑ and L‑aspartate.

Major points:

Results 3.1 – I think the author should also discuss L-enantiomers levels and/or (at least) about the D/(D+L)% ratio, which have been determined and reported in table 1. Furthermore, glycine and glutamate deserve a little attention too, in my opinion.

Response: We thank the reviewer for raising this good point. Based on the reviewer’s comments, we discussed these points as follows. First, “Large amounts of D-aspartic acid, in particular, were detected in all three major salivary glands. An earlier study found high amounts and %D, calculated…” Please see lines 361-379. Secondly, “An earlier study found high amounts of L-glutamic acid and low %D values in the following peripheral endocrine organs in rat; the thymus…“ Please see lines 392-404. Thirdly, “The results of the present study revealed that glycine levels in the salivary glands were substantially higher than …” Please see lines 455-458.

Results 3.3 – It would have been better to also analyze glutamate and glycine content in the dyalisate (interstitial fluid). The results would have important to complete the picture and perhaps would have allowed to formulate further conclusions.

Response: All data and methods on microdialysis have been removed from the revised manuscript. Please see above.

Discussion, line 329-331 – Concerning D‑aspartate levels, I think it is a pity not to have determined DDO expression levels in the three major salivary glands. I would have been possible to compare them to DAOP ones and would have probably helped explain the elevated levels of the acidic D‑amino acid in these tissues.

Response: We thank the reviewer for raising this good point. Based on the reviewer’s comments, we analyzed the DDO expression levels of gene (Figure1C) and protein (Figure 2) in the three major salivary glands and discussed these points as follows. “Functional maturation in rat salivary glands progresses over a period of 4 to 5 weeks after…” Please see lines 385-391.

Discussion, line 341-343 – Here, further analyses at different ontological stages will strengthen this hypothesis. The authors should mention it.

Response: We thank the reviewer for raising this good point. Based on the reviewer’s comments, we have now included the mention as follows; 1) “High levels of D‑aspartic acid are found in the brain and peripheral tissues during…” Please see lines 380-391;

2) “The fact that DAO was expressed…” Please see line 440-451.

Discussion, line 344-356 – So, in my understanding the principal source of D‑aspartate in these animals is from diet. I think the authors should discuss how this fit with what they proposed above (line 341-343).

Response: We thank the reviewer for raising this point, and correcting us. We apologize because our studies do not show that oral administration of high concentrations of D‑aspartate increased the concentration of D‑aspartate in endocrine organs, including salivary glands. Rather, our studies has only shown that salivary glands contain equal or greater amounts of D‑aspartate, which is reported to be highly concentrated in endocrine organs and is involved in endocrine functions. We have suggested D-aspartic acid participates in the regulation of the function of the salivary glands as an endocrine organ in the revised manuscript (lines 377-379).

Discussion– Concerning the role of DAO in controlling D‑serine levels in salivary glands, I think the authors should consider and discuss the extremely low activity reported for rat DAO on D‑serine and D‑alanine as substrates. Please see Frattini et al. (2011) FEBS J; 278:4362.

Response: We appreciate the reviewer’s suggestion and have now included such a sentence as follows; “It is, however, important to note that the substrate specificity…” Please see lines 446-448.

Discussion, line 370-371 – The authors reported DAO (as well as serine racemase) encoding gene transcription levels in salivary glands. In my opinion DAO expression (by Western blot analyses), and eventually activity, should be investigated to confirm the suggested role in regulating D‑serine levels. The authors should discuss about this.

Response: We think that the reviewer makes a reasonable request. As mentioned above, we determined protein expression levels of serine racemase, DAO, and DDO (Figure 2), in addition to gene expression level of DDO in salivary glands (Figure 1C). These assays revealed following things. First, “Expression of serine racemase was observed in salivary glands, suggesting that the serine racemase may produce endogenous D-serine there”(lines 434-436). Second, “The fact that DAO was expressed in rat salivary glands, the present finding provide further support for the view that it metabolizes endogenous D‑serine and D‑alanine in salivary glands” (lines 440-442). Third, “protein levels of DDO were lower (less than one tenth) found in the three major salivary glands in 7‑week‑old rats than in the cerebral cortex and cerebellum is in good agreement with the these earlier findings” (lines 388-391). Fourth, “The levels of serine racemase and DAO protein expression in salivary glands and cerebral cortex differed from their mRNA expression levels” (lines 483-485).

Discussion, line 380-388 – Here, measurements of glutamate and glycine concentrations would have been extremely important to substantiate the proposed notions. Without these data, I’m afraid that the reported observations are poorly consistent.

Response: As mentioned above, all data on microdialysis have been removed from the revised manuscript. In fact, we are well aware of the importance of the points made by the reviewer. We will be sure to do that in our future studies.

Discussion – The authors should mention that D‑alanine is a potent co-agonist of NMDA receptors. In the brain, its levels are very low (much lower that D‑serine ones), but here the reported levels in the interstitial fluid are even higher compared to D-serine.

Response: We thank the reviewer for raising this good point. We apologize for not being clear about this important matter in the interstitial fluid. Based on the reviewer’s comments, we added text about D‑alanine in salivary gland tissue as follows; “Both D‑serine and D-alanine acts as an endogenous coagonist for the glycine site …” Please see lines 428-433.

Discussion – Expression of NMDA receptor subunit NR2D is impressive. Could this subunit be regarded as peculiar for salivary glands? How the presence of this subunit in the tetrameric NR1/NR2D receptors may affect glutamate binding affinity? The authors think that NMDA receptors are involved in ionotropic or metabotropic signaling (or in both) in salivary glands?

Response: We thank the reviewer for raising this good point. Based on the reviewer’s comments, we added sentences about NMDA receptor subunit NR2D in discussion section. Please see lines 459-482. We, in fact, are aware of evidence showing that NMDA receptors may also operate in non-canonical or metabotropic mode in neurons and astrocytes. At this point, we can't say which type it is. However, we hope to clarify this in future studies. 

Reviewer 4 Report

The paper from Yoshikawa et al. reported the identification of free D-amino acids in salivary glands, tissue, plasma, and saliva in rats, with the aim to clarify the role of D-amino acids and their potential use as salivary biomarkers. The authors have developed a very sensitive method for the determination of D- and L-amino acids in biological samples; nevertheless, this work is a very preliminary study, it seems the optimization of a method and it must be improved. Controls and more analysis are needed.

  • which controls were used by the authors to validate HPLC analysis? the authors require the use of an enzymatic control to validate the determination of D amino acids.
  • protein levels of DAO, SR, DDO and NMDAR (Western blot or mass spectrometry) in addition to transcriptomic analysis are needed to clarify the D-amino acids contents and D-amino acids function in different tissues
  • Many references are missing in the introduction and discussion section
  • The results and the data must be better integrated in the discussion section: this section is confusing. Again, more experiments must be performed or better explanations must be provided to clarify the role of D-amino acids in these tissues (lines 357-371; 390-399; 419-424)

Author Response

The paper from Yoshikawa et al. reported the identification of free D‑amino acids in salivary glands, tissue, plasma, and saliva in rats, with the aim to clarify the role of D‑amino acids and their potential use as salivary biomarkers. The authors have developed a very sensitive method for the determination of D‑ and L‑amino acids in biological samples; nevertheless, this work is a very preliminary study, it seems the optimization of a method and it must be improved. Controls and more analysis are needed.

Response: We thank the reviewer for his/her remark.

  • which controls were used by the authors to validate HPLC analysis? the authors require the use of an enzymatic control to validate the determination of D‑amino acids.

Response: We think that the reviewer makes a reasonable request. The greatest advantage of microdialysis is that it allows quantitative changes in D‑amino acids to be measured over time. The sample volume adopted in the present study was 50 microliters. We analyzed 50-microliters samples of dialysate by HPLC after first treating them with DAO. However, it was impossible to sufficiently separate the DAO-derived contaminants in the sample solution. We tried several pretreatments aimed at removing these contaminants. As a sham test, samples that had not been treated with DAO were pretreated to remove contaminants. The amino acids were below the detection limit allowed by the HPLC analysis, however. It was also impossible to analyze the amino acids after enzymatic treatment of the dialysate. Thus, enzymatic treatment could not be performed in accordance with the guidelines of Mothet et al. 2019. It would be impossible to verify the determined D‑amino acid levels as the reviewers suggest. Therefore, all data on microdialysis have been removed from the revised manuscript. We have changed title.

  • protein levels of DAO, SR, DDO and NMDAR (Western blot or mass spectrometry) in addition to transcriptomic analysis are needed to clarify the D-amino acids contents and D-amino acids function in different tissues

Response: We think that the reviewer makes a reasonable request. We added the study of capillary electrophoresis‑based immunodetection assay (Simple Western) for determination of protein expression levels of serine racemase, D‑amino acid oxidase (DAO), D‑aspartate oxidase (DDO), and NMDA receptor subunits (Figure 2 and 5). These assays revealed following things. First, “Expression of serine racemase was observed in salivary glands, suggesting that the serine racemase may produce endogenous D-serine there”(lines 434-435). Secondly, “The fact that DAO was expressed in rat salivary glands, the present finding provide further support for the view that DAO metabolizes endogenous D‑serine and D‑alanine in salivary glands” (lines 440-442). Thirdly, “protein levels of DDO were lower (less than one tenth) found in the three major salivary glands in 7‑week‑old rats than in the cerebral cortex and cerebellum is in good agreement with the these earlier findings” (lines 388-391). Fourthly, NMDA receptor subunit proteins NR1 and NR2D were detected in all three major salivary glands (Figure 5). Fifthly, “The levels of serine racemase, DAO, NR1, and NR2D protein expression in salivary glands and cerebral cortex differed from their mRNA expression levels” (lines 483-485).

  • Many references are missing in the introduction and discussion section

Response: We appreciate the reviewer’s suggestion and have now included appropriate references in the text. Please see the introduction and discussion section.

  • The results and the data must be better integrated in the discussion section: this section is confusing. Again, more experiments must be performed or better explanations must be provided to clarify the role of D‑amino acids in these tissues (lines 357-371; 390-399; 419-424)

Response: We thank the reviewer for raising this point. We have addressed these points raised by the reviewers by shortening the text and focusing on the results gathered in the paper and the related meaning. We have followed the reviewer’s suggestion and now show gene and protein expression levels of serine racemase, DAO, DDO, and NMDA receptor subunits in salivary glands. We mainly discussed relationship between the contents of D-amino acids and expression of these enzymes, as well as that of their receptors, in salivary glands.
